# Adsorption of Pb^2+^ from Aqueous Solutions Using Novel Functionalized Corncobs via Atom Transfer Radical Polymerization

**DOI:** 10.3390/polym11101715

**Published:** 2019-10-19

**Authors:** Shanglong Chen, Wei Zhao

**Affiliations:** 1School of Chemical Engineering, China University of Mining and Technology, Xuzhou 221008, China; slchen1982@163.com; 2Jiangsu Key Laboratory of Food Resource Development and Quality Safe, Xuzhou University of Technology, Xuzhou 221018, China

**Keywords:** adsorption, ATRP, corncobs, carboxyl groups, Pb^2+^

## Abstract

The present study developed novel functionalized corncobs introducing brushes with dense and active carboxyl groups (–COOH), named MC-g-PAA, for the highly efficient adsorption of Pb^2+^ from aqueous solutions. MC-g-PAA were synthesized via atom transfer radical polymerization (ATRP) and characterized by Fourier transform infrared spectroscopy (FTIR) and scanning electron microscopy (SEM). The amount of Pb^2+^ adsorbed on MC-g-PAA by hydrolysis with t-BuOK was 2.28 times greater than that with NaOH, attributed to the larger steric effect of t-BuOK, which reduced the hydrolysis of the bromo-ester groups. The influence of different parameters including the solid/liquid ratio, working solution pH, sorption temperature, and initial concentration and sorption time on the adsorption of Pb^2+^ were investigated in detail in batch experiments. Thermodynamic studies have shown that the adsorption process was spontaneous, endothermic, and accompanied by an increase in randomness. A better fit for the isotherm data was obtained using the Langmuir model than for the other four models and the maximum amount ($q_{\max}$) of Pb^2+^ adsorbed on MC-g-PAA was 342.47 mg/g, which is 21.11 times greater when compared with that of pristine corncobs (16.22 mg/g). The adsorption of Pb^2+^ on MC-g-PAA was very fast and followed the pseudo-second-order kinetic equation with a correlation coefficient of 0.99999. This monolayer adsorption process was dominated by chemical adsorption, and may proceed according to complexation and electrostatic interactions between Pb^2+^ and the carboxylate groups. This study indicated that MC-g-PAA could be successfully used as an adsorbent for the removal of Pb^2+^ from aqueous solutions due to its excellent efficiency.

## 1. Introduction

Traditional methods for the removal of Pb^2+^ from aqueous solutions such as chemical precipitation, ion exchange, oxidation, and membrane processes may be inefficient or extremely expensive, especially when the concentration of Pb^2+^ in solution is low, resulting in an environmental issue.

Adsorption is a more effective and widely adopted method for the removal of Pb^2+^ with low concentration. The adsorbent plays a very important role in the process of adsorption. Recently, many studies have focused on the utilization of agricultural waste in the removal of Pb^2+^ [1]. The potential of biosorption to replace conventional methods has been explored in previous studies because of its faster adsorption and lower cost [2]. However, the adsorption capacity of most biosorbents in nature is still low, which is an important reason for their slow development. To improve the adsorption capacity, researchers have chemically modified biosorbents by introducing active functional groups such as carboxyl [3,4]. 

Atom transfer radical polymerization (ATRP) is a controlled radical polymerization method that has been developed in recent years [5,6,7,8], and has been extensively investigated for grafting polymers onto substrates to increase active functional groups [4,9,10,11,12]. A great many modified adsorbents via ATRP such as functionalized cotton [9,10], aminated resin [11], triethylene-tetramine grafted magnetic chitosan [12], and poly (methacrylic acid)-grafted chitosan microspheres [4] have been investigated for their adsorption of different heavy metal ions. The results [4,9,10,11,12] showed that the adsorption capacities of these modified adsorbents were significantly improved due to introducing brushes with dense and active functional groups. Recent studies on carboxylate-functionalized adsorbents [4,7,13,14] have revealed that carboxylic groups (–COOH) have a strong adsorption affinity of metal ions, which can significantly improve the adsorption capacities of the adsorbent for metal ions. Therefore, it would be ideal if corncobs could be functionalized by introducing polymer brushes with dense and active carboxyl groups (–COOH).

Corncobs are agricultural by-products with no commercial value. Dried corncobs mainly consist of cellulose (38.4%) and hemicellulose (40.7%) [15]. It has been reported that some researchers investigated the removal of Cu^2+^, Zn^2+^, Pb^2+^, Ni^2+^, and Cd^2+^ from aqueous solutions using pristine and modified corncobs as biosorbents [15,16,17,18]. The results indicated that their adsorption capacity for metal ions increased when the corncobs were modified with citric acid, phosphoric acid [15], nitric acid [16] or sulfuric acid [17]. Tan et al. [18] also reported a 1.68-time increase in the adsorption capacity for Pb^2+^ of corncobs modified with CH_3_OH and NaOH (0.1 mol/L), where the adsorption capacity of modified corncobs could reach up to 43.41 mg/g.

In this study, novel functionalized corncobs introducing brushes with dense and active carboxyl groups (–COOH), named MC-g-PAA, were synthesized as adsorbents via atom transfer radical polymerization (ATRP) with the objective to investigate in detail the influence of different parameters including the solid/liquid ratio, working solution pH, sorption temperature, and the initial concentration and sorption time on the adsorption of Pb^2+^, which is a hazardous metal ion. Thermodynamics, isotherms, and kinetics were studied using their respective theoretical models to evaluate the effectiveness of MC-g-PAA for the adsorption of Pb^2+^ from aqueous solutions. The purpose of these studies was to elucidate the adsorption mechanisms. Furthermore, the novel functionalized corncob (MC-g-PAA) surfaces were characterized by Fourier transform infrared spectroscopy (FTIR) and scanning electron microscopy (SEM).

## 2. Experimental

### 2.1. Reagents and Standards

2-bromoisobutyratebromide (BIBB, 98%), *N*,*N*,*N*,*N*,*N*-pentamethyldiethylenetriamine (PMDETA, 99%), potassium tert-butoxide (t-BuOK, 99%), 1,5-diazabicyclo [5.4.0] undec-5-ene (DBU, 98%), diisopropanolamine (DIPA, 98%), and lithium diisopropylamide (LDA, 98%) were purchased from Aldrich Chemical Company (Milwaukee, WI, USA). The standard Pb^2+^ stock solution (100 mg/L) was obtained from the National Chemical Reagent Company (Beijing, China) and diluted to the desired concentrations with deionized water. All other reagents for this study were procured from the Chinese Medicine Group Chemical Reagent Co. Ltd. (Shanghai, China) unless otherwise mentioned.

### 2.2. Preparation of the Corncobs

Corncobs were collected from the suburb of Xuzhou, China. These were thoroughly washed with deionized water, and dried at 333.15 K inside a convection oven for 24 h. Once the dried corncobs were cooled, they were crushed and sieved through a No. 50 mesh to obtain particles sized <0.3 mm. The ground corncobs were denoted as the GC.

About 50 g of GC was placed into a 1000 mL conical flask with a mixture of 250 mL ethanol and 250 mL NaOH solution (1 mol/L). The flask was shaken vigorously at a speed of 160 r/min for 1 h at 303.15 K to ensure that the extractive impurities were removed. After decantation and filtration, the treated corncobs (denoted as the TC) were washed with deionized water to remove the excess alkali and any other soluble impurities. The washing process was repeated until the pH value reached about 7.0. The TC were then dried at 333.15 K for 24 h inside a convection oven. This dried product was crushed and sieved through a No. 50 mesh.

### 2.3. Synthesis

#### 2.3.1. Immobilization of ATRP Initiators on the Corncobs

As shown in Scheme 1, immobilization of the ATRP initiators on the corncobs was prepared as follows. First, 200 g of TC were transferred to 2 L reactor system (ChemRxnHub™, Chemglass Life Sciences, Vineland, FL, USA) with 1500 mL *N*-methyl-2-pyrrolidone (NMP). The mixture of 100 mL 2-bromoisobutyratebromide (BIBB) and 100 mL NMP was added dropwise into the reactor system using a constant pressure funnel in ice cold condition after 30 min of degassing with ultrapure nitrogen. Then, the reaction mixture was stirred continuously at a speed of 80 r/min for 24 h and left for 12 h at 333.15 K under an ultrapure nitrogen atmosphere. After the reaction, the products (containing the macroinitiator corncob-Br and denoted as the MC-Br) were washed with copious amount of ethanol. Finally, MC-Br was dried at 323.15 K inside a vacuum drying oven for 24 h. These dried products were crushed and sieved through a No. 50 mesh.

#### 2.3.2. Grafting of Poly(Methyl Acrylate) on the Corncobs

MC-Br was used to start the polymerization of methyl acrylate (MA) via ATRP, where CuBr was used as the catalyst. First, 40 g of MC-Br was poured into the 2 L reactor system with 1000 mL MA. A total of 8 g CuBr and 12 mL PMDETA were added successively into the reactor systems after 10 min of continuous agitation. Then, the reactor system was degassed with ultrapure nitrogen for 30 min. The reaction mixture was stirred continuously at a speed of 80 r/min for 24 h and left for 24 h at 303.15 K under an ultrapure nitrogen atmosphere. After the reaction, the products (containing the poly(methyl acrylate) grafted corncobs and denoted as MC-g-PMA) were synthesized and washed with copious amount of methanol, acetone, tetrahydrofuran, ethanol, and deionized water, in that order [19]. Finally, MC-g-PMA was dried at 323.15 K inside a vacuum drying oven for 24 h. These dried products were crushed and sieved through a No. 50 mesh. The grafting yield was determined using the following equation:(1)$$GY\left(\%\right)=\frac{m_{2}-m_{1}}{m_{1}}\times 100,$$
where GY(%) represents the grafting yield; $m_{1}$ (g) is the mass of the dry MC-Br before grafting; and $m_{2}$ (g) is the mass of the dry products after grafting.

#### 2.3.3. Preparation of Functionalized Corncobs

The methyl acrylate of MC-g-PMA was turned into acrylic acid by hydrolysis, thus obtaining the poly(acrylic acid) grafted corncobs. About 2 g of MC-g-PMA was placed into a 100 mL conical flask with 50 mL of t-BuOK solution (0.4 mol/L). The flask was shaken vigorously in a thermostated shaker for 8 h at 160 r/min and 333.15 K. After decantation and filtration, the products (containing the poly(acrylic acid) grafted corncobs and denoted as MC-g-PAA) as the functionalized corncobs were washed with deionized water to remove excess t-BuOK and by-products of hydrolysis. The washing process was repeated until the pH value reached about 7.0. Finally, the MC-g-PAA was freeze-dried into a spongelike material inside a lyophilizer.

The hydrolysis yield was determined using the following equation:(2)$$HY\left(\%\right)=\frac{m_{4}}{m_{3}}\times 100,$$
where HY(%) represents the hydrolysis yield; $m_{3}$ (g) is the mass of dry MC-g-PMA before hydrolysis; and $m_{4}$ (g) is the mass of the dry products after hydrolysis.

Various grades of MC-g-PMA were prepared by varying the alkalis (NaOH, t-BuOK, DBU, DIPA, LDA), concentrations of alkali (0.02–0.5 mol/L), hydrolysis times (0.5–12 h), and hydrolysis temperatures (303.15–353.15 K), and optimized the best conditions of preparation with respect to a higher amount of Pb^2+^ adsorbed.

### 2.4. Characterization

Functional groups were characterized by FTIR. The FTIR spectra of TC, MC-Br, MC-g-PMA, and MC-g-PAA were recorded on a FTIR spectrometer (Nicolet iS10, Thermo Fisher Scientific, Medison, WI, USA) using the KBr dispersion method. The KBr used in this study was of pure spectroscopic grade.

In order to analyze the morphological characteristics of GC, TC, MC-Br, MC-g-PMA, and MC-g-PAA, before and after adsorption of Pb^2+^, their images were obtained by a SEM (INSPECT S50, FEI Company, Hillsboro, OR, USA) at 10 kV accelerating voltage, at 2000× magnification.

### 2.5. Pb^2+^ Solutions

Pb^2+^ working solution (500 mg/L) was prepared by dissolving lead nitrate (7.99 g) in deionized water (10 L). Other desired solutions with lower concentrations of Pb^2+^ were obtained by proper dilutions of the working solution. The working solution pH was adjusted to the working value using a small amount of hydrochloric acid or sodium hydroxide. These solutions were prepared for the adsorption experiments.

### 2.6. Adsorption Experiments

Experiments were carried out by mixing dried adsorbents with the Pb^2+^ working solution. A total of 0.10 g of MC-g-PAA was transferred to a 100 mL conical flask with 75 mL of Pb^2+^ working solution (pH 5.0). The flask was shaken vigorously in a thermostated shaker for 2 h at 160 r/min and 303.15 K to ensure that the adsorption of Pb^2+^ approached the equilibrium. Flask contents were then removed and filtered to determine the residual Pb^2+^ concentration.

One parameter was varied at a time during optimization of the batch adsorption parameters for MC-g-PAA. The effect of the MC-g-PAA solid/liquid ratio was studied in the range of 0.005–0.25 g in 75 mL of working solution. The working solution pH was varied from 1.0 to 5.0. Adsorption thermodynamics was studied at various sorption temperatures in the range of 303.15–353.15 K. Adsorption isotherms were studied at various initial Pb^2+^ concentrations ranging from 50 to 500 mg/L with the MC-g-PAA dosage of 0.10 g/(200 mL). Adsorption kinetics was studied at various sorption times in the range of 0–180 min. 

An analytical curve was made for calibration through the external standard method using standard Pb^2+^ solutions in a working range of 1–12 mg/L, which were obtained by proper dilutions of the standard Pb^2+^ stock solution (100 mg/L). The initial and equilibrium concentrations of Pb^2+^ were measured using the ContrAA 700 HR-CS AAS (Analytik Jena, Jena, Germany). All experiments were conducted three times and the mean values were presented in all cases. The removal efficiency for Pb^2+^ of MC-g-PAA was determined using the following equation [20]:(3)$$RE\left(\%\right)=\frac{c_{i}-c_{e}}{c_{i}}\times 100$$

The amount of Pb^2+^ adsorbed on MC-g-PAA was calculated according to the following equation [20]:(4)$$q_{e}=\frac{\left(c_{i}-c_{e}\right)\times V}{1000\times W},$$
where RE (%) represents the removal efficiency for Pb^2+^ of MC-g-PAA at equilibrium; $q_{e}$ (mg/g) represents the amount of Pb^2+^ uptake per unit mass of the MC-g-PAA at equilibrium; $c_{i}$ and $c_{e}$ (mg/L) are the initial and equilibrium concentrations of Pb^2+^, respectively; V (mL) is the volume of the working solution; and W (g) is the dry mass of the MC-g-PAA.

## 3. Results and Discussion

### 3.1. Synthesis

The synthesis procedure of MC-g-PAA is illustrated in Scheme 1. In brief, the ground corncobs (GC) were first treated with ethanol and NaOH in order to hydrolyze esters and remove the extractive impurities. There were many reactive hydroxyl groups (–OH) on the treated corncobs (TC). Subsequently, these hydroxyl groups (–OH) and 2-bromoisobutyratebromide (BIBB) were reacted to produce the bromo-ester groups by condensation to provide alkyl bromide initiators (MC-Br). The polymerization reaction of MA via ATRP was then performed to immobilize poly(methyl acrylate) (PMA) brushes on the MC-Br (MC-g-PMA). Finally, the prepared MC-g-PMA were functionalized through hydrolysis with alkali in order to introduce high densities of carboxyl groups (–COOH) on the corncobs (MC-g-PAA).

The grafting yield was calculated as 778.35% using Equation (1). The higher the grafting yield, the more modified groups are produced. In order to obtain dense and active carboxyl groups (–COOH), it is expected that more methyl acrylate of MC-g-PMA is turned into acrylic acid by hydrolysis. However, the bromo-ester groups fabricated from the reaction of the hydroxyl groups on the corncobs with BIBB is possibly hydrolyzed in this process, resulting in the shedding of brushes with high densities of carboxyl groups onto the corncobs. Thus, to avoid this problem, the hydrolysis conditions were optimized.

Alkalis, the concentrations of alkali, hydrolysis times, and hydrolysis temperatures were investigated. NaOH, DBU, LDA, t-BuOK, and DIPA were tested (Figure 1), with a smaller amount of Pb^2+^ adsorbed on MC-g-PAA and a higher hydrolysis yield by hydrolysis with DBU, LDA, or DIPA, indicating that poly(methyl acrylate) (PMA) brushes of MC-g-PMA were rarely hydrolyzed and could not form many carboxyl groups (–COOH) on the corncobs. The amount of Pb^2+^ adsorbed on MC-g-PAA by hydrolysis with NaOH was 121.48 mg/g, but its hydrolysis yield was the lowest, suggesting that many poly(methyl acrylate) (PMA) brushes of MC-g-PMA were hydrolyzed, but most of the poly(acrylic acid) (PAA) brushes as hydrolyzates had come off. This is because the bromo-ester groups had also been hydrolyzed in the process. The amount of Pb^2+^ adsorbed on the MC-g-PAA by hydrolysis with t-BuOK reached its maximum of up to 277.51 mg/g, which is 2.28 times greater than that with NaOH, and attributed to its higher hydrolysis yield. The hydrolysis yield with t-BuOK was 30.9%, higher than that with NaOH because the larger steric effect of t-BuOK reduced the hydrolysis of the bromo-ester groups. Therefore, the best alkali to hydrolyze MC-g-PMA is t-BuOK.

Similarly, as shown in Figure 2, Figure 3 and Figure 4, the maximum repeatable amount of Pb^2+^ adsorbed on MC-g-PAA was obtained using a concentration of 0.4 mol/L, a hydrolysis time of 8 h, and a hydrolysis temperature of 333.15 K.

### 3.2. Characterization

The FTIR spectra of TC, MC-Br, MC-g-PMA, and MC-g-PAA are shown in Figure 5. Peaks at 3417, 2904, 1640, 1378, and 1038 cm^−1^ were observed in the spectrum of TC. The broad and intense peak at 3417 cm^−1^ and the peak at 1378 cm^−1^ can be attributed to the O–H stretching and bending vibrations, respectively [18,21,22], which confirmed the presence of “free” hydroxyl groups on the TC. The peak at 2904 cm^−1^ is due to the C–H stretching vibration [18]. The peak observed at 1640 cm^−1^ was assigned to the C=O stretching vibration [18]. The presence of polysaccharide was manifested through a strong peak at 1038 cm^−1^ [18]. Compared to the spectrum of TC, there was a new peak at 1735 cm^−1^ in the spectrum of MC-Br, which can be attributed to the stretching vibration of C=O in the esters (the bromo-ester groups fabricated from the reaction of the hydroxyl groups on the corncobs with BIBB) [13,18,23,24,25]. This provides proof of successfully introducing the initiator groups on corncobs. In the spectrum of MC-g-PMA, the strong peaks that were evident at 1451 and 1735 cm^−1^ were due to the asymmetric deformation vibration of CH_3_ and the stretching vibration of C=O in the poly(methyl acrylate) [13,18,23,24,25], indicating that the dense PMA brushes were successfully grafted on the corncobs. In the spectrum of MC-g-PAA, the peaks at 1451 and 1735 cm^−1^ almost disappeared and a strong new peak at 1579 cm^−1^, due to the asymmetric stretching vibration of carboxyl groups in the poly(acrylic acid), can be observed [22,23], which indicates that the methyl acrylate of MC-g-PMA had been turned into acrylic acid by hydrolysis.

The surface morphologies of the functionalized corncobs at various stages were characterized by SEM. The SEM photographs of GC, TC, MC-Br, MC-g-PMA, and MC-g-PAA, before and after the adsorption of Pb^2+^ at 2000× magnification, are shown in Figure 6. It was observed that the surface of the GC was rough and uneven, and there were some cracks on the surface (Figure 6a). Due to the removal of some elements from GC by the mixture of ethanol and NaOH, TC had coarser and completely irregular surfaces as well as a plentiful variety of pores (Figure 6b). Some pores disappeared on the MC-Br surfaces (Figure 6c), which was due to the grafting of BIBB. This provides proof of successfully introducing the initiator groups on corncobs. The surfaces of the MC-g-PMA became smooth and some relatively close-grained polymers were clearly observed (Figure 6d), indicating that the dense PMA brushes were successfully grafted onto the corncobs. After the hydrolysis, the surfaces of the MC-g-PAA become coarse and have a good deal of variously sized pores owning to the removal of methyl groups and the shedding of some brushes with high densities of carboxyl groups from MC-g-PMA (Figure 6e). The SEM image of MC-g-PAA loaded with Pb^2+^ shows that some surfaces became uneven, but not rough, and that plenty of pores disappeared (Figure 6f), which indicates that Pb^2+^ in an aqueous solution is adsorbed on the MC-g-PAA. The introduction of brushes with high densities of carboxyl groups onto the corncobs can therefore be expected to change the surface properties and enhance the sorption of Pb^2+^.

### 3.3. Effect of Solid/Liquid Ratio

It is well known that the solid/liquid ratio greatly influences the removal efficiency for Pb^2+^ of MC-g-PAA [12,20]. The effect of the MC-g-PAA solid/liquid ratio was investigated between 0.005 and 0.25 g in 75 mL working solution and the results are shown in Figure 7.

Figure 7 shows that the removal efficiency for Pb^2+^ of MC-g-PAA was the lowest at minimum doses, and that it gradually increased as the adsorbent doses increased in the range from 0.005 to 0.20 g/(75 mL). Following this, it tended to be stable with further increases in adsorbent doses up to 0.25 g/(75 mL). Therefore, the maximum removal efficiency of Pb^2+^ was calculated as 99.5% at the solid/liquid ratio of 0.20 g/(75 mL). This can be attributed to the fact that at greater adsorbent doses, a more effective contact surface area is available for sorption.

### 3.4. Effect of Working Solution pH

It is well documented that the pH of the working solution plays an important role in adsorption of Pb^2+^ [12,13,20]. The Pb^2+^ working solution (500 mg/L) precipitates above pH 5.0, therefore, the influence of the initial pH was investigated at various pH values ranging from 1.0–5.0. The results are shown in Figure 8.

As indicated in Figure 8, a continuous increase in the adsorption capacity of Pb^2+^ on MC-g-PAA occurred in the range from 2.0 to 5.0. A small degree of sorption was observed below pH 2.0, while maximum adsorption took place at pH 5.0. Hence, the optimal pH, determined as 5.0, was used in further adsorption studies. Since Pb^2+^ and H^+^ compete for active sorption sites, the solution pH influences the sorption of Pb^2+^ on MC-g-PAA. At low pH, there is high concentration and high mobility of H^+^, which becomes adsorbed preferentially, resulting in relatively little sorption of Pb^2+^ on MC-g-PAA. As the working solution pH increases, the adsorption of Pb^2+^ also increases due to a smaller number of H^+^ and a greater number of negatively charged surface on MC-g-PAA.

### 3.5. Adsorption Thermodynamics

Thermodynamic parameters such as the free energy change ($\Delta G^{0}$), enthalpy change ($\Delta S^{0}$), and entropy change ($\Delta H^{0}$) can be estimated using thermodynamic equations [20,26]. The distribution coefficient $K_{d}$ (L/g) was determined by the following equation:(5)$$K_{d}=\frac{q_{e}}{c_{e}}$$

$\Delta G^{0}$ (kJ/mol) was calculated using the following equation:(6)$$\Delta G^{0}=-\frac{RT\ln K_{d}}{1000}$$

$\Delta S^{0}$ (J/mol/K) and $\Delta H^{0}$ (kJ/mol) were calculated using the van’t Hoff equation by plotting a graph between $\ln K_{d}$ and 1/T, where the slope gives the value of $\Delta H^{0}$ and the intercept represents $\Delta S^{0}$.
(7)$$\ln K_{d}=\frac{\Delta S^{0}}{R}-\frac{1000\times \Delta H^{0}}{RT}=-\frac{1000\times \Delta H^{0}}{R}\times \frac{1}{T}+\frac{\Delta S^{0}}{R},$$
where R (J/mol/K) represents the universal constant (8.314 J/mol/K) and T (K) is the sorption temperature.

Thermodynamic studies have been undertaken by varying the sorption temperature from 303.15−353.15 K under optimized conditions. The results are shown in Figure 9 and Figure 10.

Figure 9 shows that the adsorption capacity of Pb^2+^ on MC-g-PAA increased slowly with an increase in the sorption temperature. Figure 10 shows a graphical representation of the van’t Hoff equation as $\ln K_{d}$ against 1/T. The thermodynamic parameters for the adsorption of Pb^2+^ on MC-g-PAA were calculated using thermodynamic equations and Figure 10, and their values were recorded in Table 1.

Table 1 shows that the values of $\Delta G^{0}$ were all negative, while the values of $\Delta S^{0}$ and $\Delta H^{0}$ were positive. Negative values of $\Delta G^{0}$ indicate that the adsorption of Pb^2+^ on MC-g-PAA was a spontaneous process, which is a common observation in most metal ion uptake processes. The value of $\Delta H^{0}$ was 2.926 kJ/mol, indicating that the adsorption of Pb^2+^ on MC-g-PAA was endothermic in nature. Furthermore, the adsorption capacity of Pb^2+^ increased as the sorption temperature increased, which was in good agreement with the experimental data from Figure 9. The value of $\Delta S^{0}$ was 15.86 J/mol/K, which indicated that the adsorption of Pb^2+^ on MC-g-PAA was accompanied by an increase in randomness.

### 3.6. Adsorption Isotherms

The main objective of the isotherm is to calculate the adsorption capacity of the adsorbent, which can also be used in gaining a better understanding of the mechanism of adsorption. Isotherm studies were performed using various initial concentrations of Pb^2+^, in the range of 50–500 mg/L, with the adsorbent dosage of 0.10 g/(200 mL). The connection between the amount of Pb^2+^ adsorbed at equilibrium and the equilibrium concentration of Pb^2+^ is shown in Figure 11.

From Figure 11, it is evident that the adsorption capacity of Pb^2+^ increased abruptly as the equilibrium concentrations of Pb^2+^ increased. Following this, the adsorption capacity increased slowly after the equilibrium concentration of 23.7 mg/L. At low initial concentrations of Pb^2+^, almost all of the Pb^2+^ was adsorbed due to plenty of vacant active carboxyl groups on the MC-g-PAA, resulting in relatively low equilibrium concentrations of Pb^2+^. As the initial concentrations of Pb^2+^ increased, the equilibrium concentrations of Pb^2+^ and the adsorption capacity of Pb^2+^ also increased because of the reduction in the vacant active carboxyl groups, indicating that MC-g-PAA approached the saturated state. In this study, experimental data were analyzed using the Langmuir, Freundlich, Temkin, Dubinin–Radushkevich, and Harkins–Jura isotherm models. The linearized Langmuir isotherm model can be expressed as [20,27]:(8)$$\frac{c_{e}}{q_{e}}=\frac{1}{K_{L}\times q_{max}}+\frac{c_{e}}{q_{max}}=\frac{1}{q_{max}}\times c_{e}+\frac{1}{K_{L}\times q_{max}}$$

The dimensionless separation factor ($R_{L}$) of the Langmuir isotherm model is represented by [20,27]:(9)$$R_{L}=\frac{1}{1+K_{L}\times c_{i}},$$
where $q_{max}$ (mg/g) represents the maximum amount of Pb^2+^ adsorbed for monolayer formation on the adsorbent; $K_{L}$ (L/mg) is the Langmuir isotherm constant related to the adsorption energy; and $c_{i}$ (mg/L) is the initial concentrations of Pb^2+^. According to the $R_{L}$ values, the adsorption of Pb^2+^ from aqueous solutions using MC-g-PAA can be linear ($R_{L\;}$= 1), unfavorable ($R_{L}\;$> 1), favorable (0 <$\;R_{L}\;$< 1), and irreversible ($R_{L}\;$= 0). The values of $q_{max}$ and $K_{L}$ were calculated from the slope and intercept of the linear plot between $c_{e}/q_{e}$ and $c_{e}$.

The linearized Freundlich isotherm model is given by [20,28]:(10)$$\log q_{e}=\log K_{F}+\frac{1}{n}\times \log c_{e}=\frac{1}{n}\times \log c_{e}+\log K_{F},$$
where $K_{F}$ (mg/g) and n are constants of the Freundlich isotherm, which describe the adsorption capacity and intensity, respectively. The $K_{F}$ and n values were evaluated from the slope and intercept of the linear plot between $\log q_{e}$ and $\log c_{e}$.

The linearized Temkin isotherm model may be written as [29,30]:(11)$$q_{e}=\frac{RT}{b_{T}}\ln K_{T}+\frac{RT}{b_{T}}\ln c_{e},$$
where $b_{T}$ (J/mol) and $K_{T}$ (L/mg) are constants of the Temkin isotherm. The values of $b_{T}$ and $K_{T}$ were calculated from the slope and intercept of the linear plot between $q_{e}$ and $lnc_{e}$.

The linearized Dubinin–Radushkevich isotherm model is defined by [29,30]:(12)$$\ln q_{e}=\ln q_{max}-\beta \varepsilon^{2},$$

(13)$$\varepsilon =RT\ln\left(1+\frac{1}{c_{e}}\right)$$

The adsorption free energy (E) was determined as follows:(14)$$E=\frac{1}{\sqrt{2\beta }},$$
where ε (J/mol) is the adsorption potential energy and β is the constant of the Dubinin–Radushkevich isotherm. The values of $q_{max}$ and β were calculated from the slope and intercept of the linear plot between $\ln q_{e}$ and $\varepsilon^{2}$.

The linearized Harkins–Jura isotherm model is given by [31]:(15)$$\frac{1}{q_{e}^{2}}=\frac{B}{A}+\frac{1}{A}\times \log c_{e},$$
where A and B are constants of the Harkins–Jura isotherm. The A and B values were evaluated from the slope and intercept of the linear plot between $\frac{1}{q_{e}^{2}}$ and $\log c_{e}$.

Their correlation coefficients (R^2^) and the parameters of the five models are presented in Table 2. The linear plots of the Langmuir isotherm model is shown in Figure 12. 

It is evident from Table 2 that the correlation coefficient value of the Langmuir isotherm model (0.9994) was higher than that of other isotherm models, which clearly indicates that the Langmuir isotherm model better fitted the isotherm data. Therefore, the results show that the adsorption of Pb^2+^ on MC-g-PAA is monolayer adsorption and the distribution of carboxyl group grafted corncobs via ATRP were homogeneous. The values of $R_{L}$ were calculated to be between 0.0074 and 0.0698, indicating that the adsorption of Pb^2+^ from aqueous solutions using MC-g-PAA can be favorable within the concentration range of 50–500 mg/L. In addition, the theoretical maximum amount ($q_{max}$) of Pb^2+^ adsorbed on MC-g-PAA has been determined as 342.47 mg/g, which is in good agreement with the experimental $q_{max}$ value (341.34 mg/g) and 21.11 times greater than the $q_{max}$ value of pristine corncobs (16.22 mg/g) [18]. The results confirmed that the introduction of brushes with high densities of carboxyl groups onto the corncobs could significantly enhance the adsorption capacity of corncobs for Pb^2+^.

Table 3 shows that many modified adsorbents have been applied to the adsorption of Pb^2+^ from aqueous solutions, and some of these adsorbents have been reported to have high adsorption capacity for Pb^2+^ in recent years. Obviously, the adsorption capacity for Pb^2+^ of MC-g-PAA was higher than those of other adsorbents and much higher than those of the adsorbents with corncobs as the raw materials. Hence, it can be concluded that the MC-g-PAA synthesized via ATRP herein has strong competitive advantages for the adsorption of Pb^2+^ from aqueous solutions.

### 3.7. Adsorption Kinetics

The adsorption rate is one of the most important features of the adsorbent; therefore, the effect of sorption time was studied between 0 and 180 min in this study. The results of Pb^2+^ adsorption on MC-g-PAA are shown in Figure 13. It can be seen that adsorption is a very fast process. The adsorption capacity of Pb^2+^ on MC-g-PAA increased rapidly in the first 15 min, contributing to 68.82% at 1 min and 98.49% of the overall adsorption capacity (324.68 mg/g) at 15 min. The continuing increase was slower and approached the adsorption equilibrium after 15 min. The rapid adsorption rate reflects the good accessibility of Pb^2+^ binding sites on MC-g-PAA, which is beneficial in practical application, since it results in a reduction of reactor volumes and time.

In order to evaluate the kinetic mechanism, the experimental kinetic adsorption data were analyzed by using six models: the pseudo-first-order, pseudo-second-order, intra-particle diffusion, Elovich, liquid-film diffusion, and Avrami-fractional kinetic models. The linearized pseudo-first-order kinetic model can be written as [41]:(16)$$\frac{1}{q_{t}}=\frac{k_{1}}{q_{e}\times t}+\frac{1}{q_{e}}=\frac{k_{1}}{q_{e}}\times \frac{1}{t}+\frac{1}{q_{e}},$$
where t (min) represents the sorption time; $q_{t}$ (mg/g) is the amount of Pb^2+^ adsorbed per unit mass of the adsorbent at t min; and $k_{1}$ (min^−1^) is the adsorption rate constant of the pseudo-first-order model. The $q_{e}$ and $k_{1}$ values were determined from the slope and intercept of the linear plot between $1/q_{t}$ and 1/t.

The linearized pseudo-second-order kinetic model is given by [2,41]:(17)$$\frac{t}{q_{t}}=\frac{1}{k_{2}\times q_{e}{}^{2}}+\frac{1}{q_{e}}\times t=\frac{1}{q_{e}}\times t+\frac{1}{k_{2}\times q_{e}{}^{2}},$$
where $k_{2}$ (g/mg/min) represents the adsorption rate constant of the pseudo-second-order model. The values of $q_{e}$ and $k_{2}$ were evaluated from the slope and intercept of the linear plot between $t/q_{t}$ and t.

The initial adsorption rate (h) was determined as follows [41]:(18)$$h=k_{2}\times q_{e}{}^{2}$$

The linearized intra-particle diffusion kinetic model is represented by [30,42,43,44]:(19)$$q_{t}=k_{F}\times t^{0.5}+C,$$
where $k_{F}$ (mg/g/min^0.5^) represents the intra-particle diffusion rate constant and C is the intercept related to the thickness of the boundary layer. The values of $k_{F}$ and C were evaluated from the slope and intercept of the linear plot between $q_{t}$ and $t^{0.5}$.

The linearized Elovich kinetic model is defined by [44,45]:(20)$$q_{t}=\frac{1}{\beta }ln\left(\alpha \beta \right)+\frac{1}{\beta }\ln t=\frac{1}{\beta }\ln t+\frac{1}{\beta }ln\left(\alpha \beta \right),$$
where α (min/mg/g) and β (g/mg) are constants of the Elovich model. The α and β values were estimated from the slope and intercept of the linear plot between $q_{t}$ and lnt.

The linearized Avrami-fractional kinetic model is represented by [43]:(21)$$\ln\left(-\ln\left(1-\frac{q_{t}}{q_{e}}\right)\right)=n_{AV}\times lnt+n_{AV}\times lnK_{AV},$$
where $n_{AV}$ and $K_{AV}$ are constants of the Avrami-fractional model. The $n_{AV}$ and $K_{AV}$ values were estimated from the slope and intercept of the linear plot between $\ln\left(-\ln\left(1-\frac{q_{t}}{q_{e}}\right)\right)$ and lnt.

Kinetic parameters for the adsorption of Pb^2+^ on MC-g-PAA and the correlation coefficients (R^2^) of these kinetic models are given in Table 4.

Table 4 shows that the correlation coefficient value of the pseudo-second-order kinetic model (0.99999) was higher than that of other kinetic models. The linear plot of the pseudo-second-order kinetic model is shown in Figure 14.

Obviously, the pseudo-second-order kinetic model provides a better fit for experimental kinetic adsorption data than that of the other five models, which suggests that the adsorption process is dominated by chemical adsorption involving complexation and electrostatic interactions [4].

The $k_{2}$, h, and $q_{e}$ values of the pseudo-second-order model were found to be 0.00561 g/mg/min, 591.72 mg/g/min, and 324.68 mg/g, respectively. The initial adsorption rate (h) was extremely high, indicating that the adsorption of Pb^2+^ on MC-g-PAA is a rapid process. The theoretical $q_{e}$ value was in good agreement with the experimental $q_{e}$ value (323.66 mg/g).

## 4. Conclusions

The brushes with dense and active carboxyl groups (–COOH) was successfully introduced onto the corncobs via the ATRP technique, obtaining functionalized corncobs (MC-g-PAA) as novel adsorbents, which were characterized by FTIR and SEM. The solid/liquid ratio and solution pH had strong effects on the adsorption of Pb^2+^, and their optimum values were determined as 0.20 g/(75 mL) and 5.0, respectively. Thermodynamic studies showed that the adsorption capacity of Pb^2+^ on MC-g-PAA increased slowly with the increase of sorption temperature. Furthermore, this adsorption process was spontaneous, endothermic, and accompanied by an increase in randomness. The Langmuir model provided a better fit for the isotherm data than the Freundlich model and the maximum amount ($q_{max}$) of Pb^2+^ adsorbed on MC-g-PAA was 342.47 mg/g, which was 21.11 times greater than the $q_{max}$ of the pristine corncobs (16.22 mg/g). Kinetic studies showed that adsorption was a very fast process, which approached an equilibrium after 15 min, and that the experimental data had the best agreement with the pseudo-second-order equation. The monolayer adsorption process of Pb^2+^ on MC-g-PAA was dominated by chemical adsorption, and may proceed according to complexation and electrostatic interactions between Pb^2+^ and the carboxylate groups.

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
