# Peer review of "Adsorption of Pb2+ from Aqueous Solutions Using Novel Functionalized Corncobs via Atom Transfer Radical Polymerization"

_polymers, 2019, doi:10.3390/polym11101715_

Round 1

Reviewer 1 Report

Dear Authors,

Please find attached my comments on your article. It could be interesting but major revision should be made and some parts of manuscript needs to be explained or improved.

Reviewer 2 Report

This is an interesting dealing with the adsorption of Pb2+ from aqueous solution using functionalized corncob via ATRP technique. Authors showed that maximum amount of adsorbed Pb2+ ions significantly increases after ATRP treatment. Nevertheless, some aspects must be taken into account before being considered for publication. My comments are listed below:

First part of Abstract is not clearly written. Better explanation is needed. Equations 1, 2, 3 are incorrect: if results will be given in % than in all equations missed x100 P2/L51-52: ʺIt would be ideal if corncobs can be functionalized by polymer brushes with dense and active carboxyl groups (-COOH).ʺ Please explain why it would be ideal? Did authors use solid lead nitrate or liquid standard lead solution (100 mg/L) for preparing working solutions (P2/L75-76; P4/L139)? P3/L105: What is PMDETA? Part 3.1. Synthesis is not clearly written. Please better explain that part. P6/L178: 778.35% - is it possible? P6/L211: From the text it can be expected that Figure 5 will include SEM images of all materials (GC, TC, MC-Br, MC-g-PMA and MC-g-PAA) before and after lead adsorption. Please check. P8/L237: 0.005 or 0.05 g? Please add the proper references to support facts in 3.3 and 3.4 parts. Optimal mass is founded to be 0.2g/75ml but authors used mass of 0.1g/200ml for adsorption isotherm study. Please provide some explanation why did not used optimal mass? Please provide the proper references for all isotherm models. Figure 10 is incorrect due to equilibrium concentration cannot affect adsorption process. P 16/ L417-418: ʺThe brushes with dense and active carboxyl groups (-COOH) had been successfully introduced onto the corncobs via ATRP technique, obtaining functionalized corncobs (MC-g-PAA) as novel biosorbentsʺ How authors concluded that active carboxyl groups (-COOH) had been successfully introdused? Please add some fact or provide more results for which will confirme that claim. There are a lot of grammar mistakes. Please check English in whole Manuscript

Round 2

Reviewer 1 Report

Dear Authors,

I am glad to see your improved manuscript. The manuscript has been significantly improved and now is appropriate to be published in Polymers.

Author Response

Thank you!

Reviewer 2 Report

Please describe how the grafting yield was 778.35%, if it is mistake please correct (P5/L193) You did not provide explanation about Figure 11 (previously was Figure 10)
